# Eco-Friendly Water-Processable Polyimide Binders with High Adhesion to Silicon Anodes for Lithium-Ion Batteries

**DOI:** 10.3390/nano11123164

**Published:** 2021-11-23

**Authors:** Yujin So, Hyeon-Su Bae, Yi Young Kang, Ji Yun Chung, No Kyun Park, Jinsoo Kim, Hee-Tae Jung, Jong Chan Won, Myung-Hyun Ryou, Yun Ho Kim

**Affiliations:** 1Advanced Materials Division, Korea Research Institute of Chemical Technology (KRICT), Daejeon 34114, Korea; soyujin@krict.re.kr (Y.S.); yykang@krict.re.kr (Y.Y.K.); jjyjjy@krict.re.kr (J.Y.C.); nkpark@krict.re.kr (N.K.P.); jinsoo@krict.re.kr (J.K.); 2Department of Chemical and Biomolecular Engineering, Korea Advanced Institute of Science and Technology (KAIST), Daejeon 34141, Korea; heetae@kaist.ac.kr; 3Department of Chemical and Biological Engineering, Hanbat National University, Daejeon 34158, Korea; hsbae1234@gmail.com; 4Korea Research Institute of Chemical Technology (KRICT) School, University of Science and Technology, Daejeon 34113, Korea

**Keywords:** lithium-ion batteries, silicon anodes, binder, polyimide, water-processable

## Abstract

Silicon is an attractive anode material for lithium-ion batteries (LIBs) because of its natural abundance and excellent theoretical energy density. However, Si-based electrodes are difficult to commercialize because of their significant volume changes during lithiation that can result in mechanical damage. To overcome this limitation, we synthesized an eco-friendly water-soluble polyimide (W-PI) precursor, poly(amic acid) salt (W-PAmAS), as a binder for Si anodes via a simple one-step process using water as a solvent. Using the W-PAmAS binder, a composite Si electrode was achieved by low-temperature processing at 150 °C. The adhesion between the electrode components was further enhanced by introducing 3,5-diaminobenzoic acid, which contains free carboxylic acid (–COOH) groups in the W-PAmAS backbone. The –COOH of the W-PI binder chemically interacts with the surface of Si nanoparticles (SiNPs) by forming ester bonds, which efficiently bond the SiNPs, even during severe volume changes. The Si anode with W-PI binder showed improved electrochemical performance with a high capacity of 2061 mAh g^−1^ and excellent cyclability of 1883 mAh g^−1^ after 200 cycles at 1200 mA g^−1^. Therefore, W-PI can be used as a highly effective polymeric binder in Si-based high-capacity LIBs.

## 1. Introduction

Lithium-ion batteries (LIBs) comprising Li ions as electron carriers have dominated the energy storage market owing to their high energy density, high operating voltage, and low self-discharge properties [1,2]. In recent years, electric vehicles (EVs) and energy storage systems (ESSs) have been increasingly developed as alternatives to fossil fuel use to reduce greenhouse gas emissions. Increasing the energy density of LIBs is highly desirable for the successful implementation of EVs and ESSs, as commercial LIBs are approaching the theoretical limit of the commonly used materials. In general, commercial LIBs consist of carbon-based anodes (e.g., graphite with a theoretical capacity of 372 mAh g^−1^) and Li metal transition oxide cathodes (e.g., Li cobalt oxide, LiCoO_2_). To increase the energy density of LIBs, Si has been extensively studied as an anode material because of its exceptionally high theoretical energy density (4200 mAh g^−1^), low reduction potential (0.2 V vs. Li/Li^+^), and low cost [3,4,5,6,7]. Si-active materials undergo an alloying/dealloying process (4.4Li^+^ + 4.4e^−^+ Si ↔ Li_4_._4_Si) [8,9], and their theoretical capacity can reach 10 times that of graphite. However, Si-based electrodes undergo a volume change of up to 300% during lithiation (alloying). The repeated lithiation/delithiation processes during charging/discharging can cause enormous mechanical stress on the Si electrodes, which can lead to fracturing of the Si-active materials and their delamination from the Cu current collector [10]. This results in a loss of electrical contact, an increase in the internal resistance, and loss of the active material, which seriously degrades the cycling performance of the Si electrode.

Typically, anode electrodes consist of active materials, conductive additives (e.g., carbon nanopowder), and polymeric binders. Therefore, researchers have overcome Si anode degradation by: (1) structural modification of Si-active materials [11,12]; (2) surface chemistry modification of Si-active materials [13]; (3) use of functional conductive additives [14]; and (4) polymeric binder modification [15,16]. Considering efficient mass production and cost competitiveness, the use of highly functional polymeric binder materials is believed to be the most effective approach for improving the electrochemical performance of Si anodes.

The primary role of the polymeric binder is to physically connect the active materials and conductive additives in the electrode composite and adhere it to the current collector [17]. In the case of graphite based on the intercalation mechanism, the role of polymeric binders has been overlooked due to a small volume change (9.4%) upon lithiation (LiC_6_) [18]. For example, poly(vinylidene fluoride) (PVDF) has been commonly used as a polymeric binder for commercial LIB electrodes. However, PVDF cannot withstand the mechanical stress in the Si anodes and thus cannot be used for Si anodes [6,19,20,21]. Thus, polymeric binders have been developed with excellent mechanical strength, which can withstand large volume changes through interfacial interactions (e.g., hydrogen bonding) with Si-active materials [22,23,24,25,26]. For example, Lee et al. [24] prepared a polyaniline-blended poly(acrylic acid) (PAA) binder, which showed remarkable electrochemical performance. The polyaniline component provides acid–base interactions, while PAA forms hydrogen bonds with the Si electrode. Zhao et al. [25] developed poly(1-pyrene-methyl methacrylate-co-methacrylic acid) as a conductive binder, which has high conductivity and adhesion owing to the methacrylic acid and pyrene in the copolymer structure. Thus, stable first-cycle efficiency and long-term cycling performance were obtained. Ling et al. [26] applied naturally abundant gum arabic polymer as a dual-function binder; the hydroxyl groups of the polysaccharides provide strong binding to Si. The long glycoprotein chains improved the mechanical tolerance to the volume expansion of the Si electrodes. However, the thermal stability and mechanical and adhesive properties of Si electrodes still require further improvement to provide a sufficiently long lifetime under the harsh operating conditions of LIBs.

Considering this background, we hypothesized that polyimides (PIs) could be used as polymeric binders with good mechanical properties, chemical resistance, thermal stability, and moisture resistance [27]. In addition, the properties of PIs can be easily controlled by selecting monomers with desirable functional moieties. However, highly polar aprotic organic solvents such as dimethylacetamide, dimethylformamide, and N-methyl-2-pyrrolidone are required for the polymerization of poly(amic acid) (PAmA). Because these organic solvents are harmful to the environment and expensive [28], their use in the industry has been severely restricted in recent years. The use of environmentally friendly materials is strongly recommended in the secondary battery and EV industries. To overcome this drawback, we recently developed an environmentally friendly water-based synthesis method for PIs [29,30]. Water-soluble poly(amic acid) salt (W-PAmAS) can be synthesized in water via a one-step process by adding an organic base (1,2-dimethylimidazole, DMIZ) (Figure 1a). Another advantage of this technique is that W-PAmAS can lower the imidization temperature compared to conventional organic-based PAmA. Even at a low temperature of 150 °C, W-PAmAS showed an imidization degree of 91.0%. These results show that W-PI, with excellent mechanical and thermal properties, can be used as a Si anode binder. In addition, carboxylic acid moieties such as 3,5-diaminobenzoic acid (DABA) were introduced into the W-PI binder to enhance the binding force by forming covalent bonds with the Si anode nanoparticles (SiNPs). The molecular structure of W-PI obtained from W-PAmAS was carefully analyzed and systematically compared with PAmAS synthesized using an organic solvent (O-PAmAS). The chemical structures of W-PAmAS and O-PAmAs were characterized by proton nuclear magnetic resonance (^1^H NMR) and attenuated total reflectance Fourier transform infrared spectroscopy (ATR-FTIR). The electrochemical properties, such as the cycle performance and rate capability of the W-PAmAS binder-based Si anodes, were evaluated. The effects of the DABA moiety in the W-PI binder on the electrochemical and adhesive properties were systematically evaluated. The physical and adhesive properties of Si anodes with the W-PI binder were investigated by X-ray photoelectron spectroscopy (XPS), a surface and interfacial cutting analysis system (SAICAS), and scanning electron microscopy (SEM).

## 2. Materials and Methods

### 2.1. Materials

*p*-phenylenediamine (pPDA), 3,5-diaminobenzoic acid (3,5-DABA), and 3,3,4,4-Biphenyltetracarboxylic dianhydride (s-BPDA) were purchased from Tokyo Chemical Industry Co., Ltd. (Tokyo, Japan) and vacuum dried at 80 °C, 180 °C, and 250 °C, respectively. DMIZ was purchased from Sigma-Aldrich (Korea) and used as received. SiNPs (30–50 nm, >98%, Nanostructured & Amorphous Materials, Inc., Katy, TX, USA), Super-P (TIMCAL Ltd., Co., Bodio, Switzerland), Li metal (Honjo Metal Co., Ltd., Osaka, Japan, thickness = 200 μm), and poly(acrylic acid) (M_w_ = 450,000, Sigma-Aldrich, St. Louis, MO, USA) were purchased and used without further purification. A mixture of 1 M lithium hexafluorophosphate (LiPF_6_) in ethylene carbonate/ethyl methyl carbonate (3/7) containing 5 wt% fluoroethylene carbonate (FEC; Enchem Co., Ltd. Chungbuk, Korea) was used as liquid electrolytes. A microporous polyethylene separator (ND420, thickness = 20 μm, porosity = 40%, Asahi Kasei E-materials, Tokyo, Japan) was used as the separator.

### 2.2. Synthesis of Water-Soluble Poly(Amic Acid) Salts (W-PAmAS)

W-PAmAS was synthesized using 100 mL three-neck round-bottom flasks equipped with a mechanical stirrer and heating system under a N_2_ atmosphere. The molecular ratio of DABA in W-PAmAS-#, where # indicates the molecular percentage of DABA in W-PAmAS, was varied from 0% to 30% at 10% intervals to optimize the electrochemical and adhesive properties of W-PAmAS. For example, the optimized composition in this study was W-PAmAS-30 with a 30% molar ratio of DABA. The W-PAmAS synthesis procedure was as follows (Figure 1a): DMIZ (25 mmol) and diamine (10 mmol, pPDA/3,5-DABA = 7/3 by mol.%) were added to deionized (DI) water, and the mixture was stirred for 1 h at 25 °C. After DMIZ and diamine were dissolved entirely, BPDA (10 mmol) was added to the resulting solution. The mixture was heated and stirred at 70 °C for 18 h. After polymerization, the viscous solution with orange-brown color was obtained, and the resulting W-PAmAS-30 aqueous solution had an inherent viscosity of 0.46 dL g^−1^.

### 2.3. Characterizations

^1^H NMR spectra were obtained using a Bruker Avance-500 MHz FT-NMR spectrometer (Bruker, Billerica, MA, USA). W-PAmAS-0 and W-PAmAS-30 were dissolved in dimethyl sulfoxide (DMSO-d_6_) after drying at 25 °C for 12 h. Attenuated total reflectance Fourier transform infrared (ATR-FTIR) spectra were collected using a Bruker ALPHA-FT-IR spectrometer over the wavenumber range of 650–4000 cm^−1^. Heat-treated W-PAmAS (denoted as W-PI) samples were prepared by casting W-PAmAS solution onto a glass plate, followed by drying at 25 °C for 12 h. To evaluate the effect of the degree of imidization, two W-PI samples were prepared by heating W-PAmAS at different temperatures of 150 °C or 350 °C for 30 min. XPS (Kratos Axis-Supra, Manchester, UK) was performed with monochromatic AlKα radiation at 10^−9^–10^−8^ Torr.

### 2.4. Preparation of Silicon Electrodes

A water-based Si electrode slurry consisting of SiNPs/conductive material (Super-P)/polymeric binders (PAA, W-PAmAS-0, and W-PAmAS-30) (60/20/20 by wt%) was coated on a copper foil (11 μm, Iljin Materials, Korea) using a gap-controlled doctor blade and dried at 80 °C for 1 h in a convection oven. The dried Si electrodes were prepared using a gap-controlled roll press (CLP-2025, Creative & Innovative Systems, Daegu, Korea). The thickness of the electrode was adjusted using a roll press, and the electrode was cut into disks with a diameter of 12 mm and heated at 150 °C for 6 h under vacuum. The loading level of the fabricated Si electrode was kept at 0.5 mg cm^−2^ (thickness = 11 μm, density = 0.45 g cm^−3^), regardless of the binder type.

### 2.5. Cell Assembly

CR2032 Li-metal half-cells were used to evaluate the electrochemical properties of the Si electrodes. Coin cells were assembled in an argon-filled glove box with a dew point of −80 °C. Polyethylene separators (ND420) were sandwiched between the Li metal and Si electrode, where the Li metal was used as the counter and reference electrodes and the Si electrode was used as the working electrode. A total of 600 μL of liquid electrolyte was used for coin cell assembly.

### 2.6. Electrochemical Measurements

The assembled CR2032 half-cells (Si/Li metal) were aged for 12 h at 25 °C and cycled in a constant current mode over the voltage range 0.005–2.0 V vs. Li/Li^+^ (at a rate of 200 mA g^−1^ at 25 °C) using a charge/discharge tester (PNE Solution, Republic of Korea). This process is referred to as precycling. After precycling, the CR2032 half-cells (Si/Li metal) were cycled for 200 cycles in the same voltage range at a rate of 1200 mA g^−1^. Furthermore, the rate capability of the Si electrode was evaluated by increasing the discharging (dealloying) current density up to 6.0 A g^−1^ (1.2, 2.4, 3.6, 4.8, and 6.0 A g^−1^) while maintaining the charging (alloying) current density at 0.06 A g^−1^.

### 2.7. Surface and Interfacial Cutting Analysis System (SAICAS) Measurements

The interfacial adhesion strength between the Si electrode and Cu collector was measured using SAICAS (Daipla Wintes Co., Ltd., Saitama, Japan). A boron nitride blade (width = 1 mm, rake angle = 20°) was moved horizontally at 0.1 μm s^−1^ with a vertical cutting force of 0.2 N. When a blade was in contact with the Cu collector, the electrode coating layer was delaminated. The adhesion strength of the electrodes was calculated by measuring the horizontal force during peeling at a constant rate.

### 2.8. Morphological Analysis

Surface and cross-sectional images of the Si electrodes before and after cycling were obtained using field-emission scanning electron microscopy (FE-SEM, Philips XL30S FEG, FEI Co., Hillsboro, OR, USA) to investigate the morphological changes of the Si electrodes after cycling. Cross-sections of the Si electrodes were cut using focused ion beam milling (Quanta, FEI Co., Hillsboro, OR, USA).

## 3. Results and Discussion

We prepared Si electrodes using W-PI as the polymeric binder. The detailed synthesis of W-PAmAS as a precursor of W-PI, the imidization process of W-PI, and the interaction between W-PI and Si-active materials are illustrated in Figure 1. W-PAmAS-30 was synthesized using BPDA, pPDA (70 mol.%), DABA (30 mol.%), and DMIZ as the organic base in DI water (Figure 1a). The BPDA-pPDA-based PI backbone has high thermal stability and excellent mechanical properties owing to its rigid aromatic backbone [31]. To enhance the intermolecular interactions between the active materials and polymeric binders, DABA-containing carboxylic acid groups were introduced as the comonomer. The free carboxylic acid group is transformed into carboxylate (–COO–) in aqueous solutions, which strongly interacts with SiNPs via a physical or chemical reaction [32,33]. The presence of DMIZ in W-PAmAS promotes polymerization in DI water by forming an ammonium salt with the carboxylate group of W-PAmA and acts as an organic base catalyst that decreases the thermal imidization temperature [34,35].

To fabricate the Si electrodes, W-PAmAS-30 was mixed with SiNPs and a conductive agent (Super-P), and the slurry was cast on Cu foil. The W-PAmAS-30 of the cast slurry was imidized by thermal treatment at 150 °C for 6 h, through which W-PAmAS-30 was converted to W-PI-30. Because of the carboxylic groups of DABA, W-PI-30 strongly interacts with the hydroxyl moieties of SiNPs, resulting in strong physical adhesion (Figure 1b). The effect of the W-PI-30 polymeric binders on the physical and electrochemical properties of the Si electrodes is described in detail in the following sections.

The chemical structures of W-PAmAS-0 and W-PAmAS-30 containing different amounts of DABA were confirmed by ^1^H NMR analysis using dimethyl sulfoxide (DMSO-d_6_) as the solvent. As shown in the ^1^H NMR spectrum of W-PAmAS-0 and W-PAmAS-30 (Figure 2a), the typical aromatic protons of dianhydride and diamine of PAmA were broadly shown at 7.2–8.7 ppm, denoted by “a–c” and “e–f”. The chemical shifts for the amide proton (–NH–) and the protonated amine of DMIZ appeared at 10.5–11.3 ppm, while the typical proton of carboxylic acid (–COOH) in PAmA was observed at 13.2 ppm in the ^1^H NMR spectra of O-PAmA-0 and O-PAmA-30 (Appendix A). The results indicated that 2.5 molar equivalents of DMIZ enabled polymerization by forming a salt between carboxylate anions and imidazolium cations, which was water soluble. Therefore, a new chemical shift was observed when the DABA group was introduced into the W-PAmAS-30 backbone to form a copolymer, as shown in the spectrum of W-PAmAS-30 (Figure 2a). The peak related to the aromatic proton of BPDA was divided into two at ~8 ppm, denoted by “a–c” and “l–n”, while the amide proton peak was also split into two at ~11 ppm, denoted by “d” and “o”. The –COOH in DABA also formed a salt with DMIZ, as evidenced by a weak protonated amine peak at 10.9 ppm, denoted by “s”. This is distinguished from the protonated amine peak of DMIZ at 10.65 ppm, denoted by “g”, which interacts as the salt form with the typical –COOH group of W-PAmAS. In addition, when W-PI-# was prepared by increasing the DABA content by 10% steps, the intensity of the amide proton peak increased at 11.1 ppm, denoted by “o”, resulting from the proportional increase in DABA content of the W-PAmAS-# samples (Appendix A). This clearly shows that the DABA group was introduced into W-PAmAS, as desired.

The presence of DABA introduced into W-PAmAS-# was confirmed by FT-IR analysis (Figure 2b). The characteristic O–H stretching peak of –COOH at 3620 cm^−1^ was only observed for the spectrum of W-PI-30 containing 30 mol.% DABA. To completely exclude unconverted carboxylic acid of W-PAmAS-#, both W-PI-0 and W-PI-30 were fully imidized by heat treatment at 350 °C. In the spectrum of W-PI-0, the characteristic peak of –COOH from DABA was not observed. However, the peak corresponding to the O–H stretching of –COOH was observed at 3620 cm^−1^ in the spectrum of W-PI-30, despite complete imidization. This characteristic peak originates from the –COOH moiety in the DABA group, and it was also observed for W-PI-10 and W-PI-20 (Appendix A). Based on the NMR and FT-IR results, it was confirmed that the DABA group was incorporated into W-PI-30 at the desired ratio.

We further investigated the effect of the thermal annealing temperature on the degree of imidization of W-PAmAS-30. In particular, the possibility of a low-temperature process is significant for LIB manufacturing in the case of PI materials that require heat treatment for imidization. Two different W-PI-30 samples thermally treated at 150 °C or 350 °C showed similar FT-IR results (Figure 2c). At the lower temperature of 150 °C, W-PAmAS-30 showed an imidization degree of 95.0%. It was confirmed that W-PI-30 was sufficiently imidized by heat treatment at 150 °C in the presence of DMIZ, which can act as an organic base catalyst [36]. The same trend was observed regardless of the DABA content (Appendix A). The low-temperature imidization properties of W-PAmAS have already been reported in our previous study [30]. Therefore, the physical properties of W-PI processed at low temperatures are equivalent to those of the existing O-PI binder. Considering that the Cu current collector is easily oxidized when exposed to high temperatures in air, W-PI-30 treated at 150 °C is a highly suitable polymeric binder for Si electrodes to minimize Cu oxidation.

To confirm the chemical bonding between W-PI-30 and SiNPs, the chemical composition of the Si electrodes heat treated at 150 °C using W-PI-30 as a polymeric binder, W-PI-30/SiNP, was investigated using XPS analysis. For comparison, the W-PI-30 polymer without SiNPs was evaluated as a reference. The deconvolutions of the C 1s spectra for W-PI-30 and W-PI-30/SiNP are shown in Figure 3. Both W-PI-30 and W-PI-30/SiNP showed distinct imide peaks at 284.7, 285.8, 288.2, and 290.5 eV assigned to C–C/C–H, C–N, C=O, and π–π binding, respectively [37,38]. In comparison, the additional peak at 289.1 eV assigned to covalent ester groups (–COOR–) was detected for W-PI-30/SiNP, implying that the W-PI-30 polymeric binder and SiNPs were bound through a covalent ester group formed via the chemical reaction between –COOH groups in W-PI-30 and the Si–OH bonds on the Si surface [32,33]. Because of the air contamination, the surfaces of the SiNPs were covered with oxygen moieties, such as SiOx (102.6 eV in the Si 2p spectra, Appendix A), Si–O, Si–O–Si, and Si–OH (531.6, 532.3, and 533.4 eV, respectively, in the O 1s spectra, Appendix A) [39,40]. These results imply that W-PI-30 binder can provide strong adhesion between active materials and other components in Si electrodes.

The electrochemical properties, such as the cycle performance and rate capability of the SiNPs, were investigated (Figure 4). For comparison, Si electrodes using PAA, W-PI-0, and W-PI-30 as binders were prepared (PAA/SiNP, W-PI-0/SiNP, and W-PI-30/SiNP, respectively). All samples were heat treated at 150 °C for 6 h after coating the electrode slurry onto the Cu foil. Precycling was performed that was performed in a constant current mode at a rate of 200 mA g^−1^ before cycling test, precycling curves of Si anodes with W-PI binders are showed in Appendix A. Si electrodes using the W-PI polymeric binder series (W-PI-0 and W-PI-30) showed significantly improved cycle performance compared to PAA/SiNP (Figure 4a). The unusual capacity increases were observed over multiple experiments, confirming that the results are not caused by technical artifacts. Although underlying mechanisms remain to be solved, this capacity increase is ascribed to the synergistic effect of the binder backbone and the electrolyte additive [19]. Furthermore, PAA/SiNP, W-PI-0/SiNP, and W-PI-30/SiNP maintained 77.3%, 83.0%, and 91.3% of the initial discharge capacity after 200 cycles, respectively, indicating a 107.4% and 118.1% improvement for W-PI-0/SiNP and W-PI-30/SiNP, respectively, compared to PAA/SiNP. In addition, compared with other reported polymer binders, Si anode with W-PI-30 binder shows high capacity retention of 91.3% even at relatively high current rates [4,20,22,23,24,25,26,41,42,43,44]. Cycle performances of Si-based anodes with different polymer binders are summarized into the Appendix A. Because PAA is considered a promising polymeric binder for Si-active materials, it can be inferred that the W-PI polymeric binder series offers competitive results.

Both W-PI-0/SiNP and W-PI-30/SiNP electrodes showed a more noticeable improvement in the rate capability performance than PAA/SiNP (Figure 4b). PAA/SiNP maintained a similar rate capability for discharging up to 2.4 A g^−1^ compared to W-PI-30/SiNP. However, W-PI-0/SiNP showed the lowest discharge capacity for discharging up to 2.4 A g^−1^. In particular, PAA/SiNP dropped to zero discharge capacity (0.3 mAh g^−1^) at 3.6 A g^−1^ and recovered to 1598.1 mAh g^−1^, which is similar to that of W-PI-30/SiNP (1552.5 mAh g^−1^). In general, the rate capability of single battery cell is governed by a combination of permanent electrochemical composition loss and the kinetic properties of the electrodes. The recovery of the discharge capacity of PAA/SiNP in the secondary 1.2 A g^−1^ condition after the 50th cycle implies that the kinetic behavior dominated the rate capability of PAA/SiNP. Thus, the internal resistance of PAA/SiNP was higher than that of the other SiNP systems. Moreover, W-PI-0/SiNP and W-PI-30/SiNP had similar cycle performances (Figure 4a), but showed deteriorated rate capabilities compared to PAA (Figure 4b). The difference between W-PI-0 and W-PI-30 is the presence of DABA monomers that form –COOH in the polymeric binder backbone. Forming covalent bonds on the surface of SiNPs reduces the volume change of W-PI-30/SiNP, thereby reducing active material loss and liquid electrolyte loss and maintaining physical contact between the Si-active materials.

The morphological changes in the Si electrodes after cycling were evaluated using SEM (Figure 5). Before electrochemical cycling (denoted as pristine), PAA/SiNP and W-PI-30/SiNP had similar morphology (Figure 5a,b). After precycling, the W-PI-30/SiNPs had no cracks, whereas many deep cracks were observed in PAA/SiNP (Figure 5c,d). After 20 cycles, as shown in Figure 4a, the surface and cross-sectional morphologies of PAA/SiNP and W-PI-30/SiNP were also observed using SEM (Figure 5g–j). The surface cracks observed in the PAA/SiNPs were larger and deeper than those in the precycled case. Furthermore, the thickness of PAA/SiNP (~7.4 μm) was much larger than that of W-PI-30/SiNP (~4.8 μm) (Figure 5g,h). As discussed in the introduction, Si electrodes are accompanied by enormous volume changes during repeated alloying/dealloying cycles. If the mechanical toughness and adhesion properties of the polymeric binder are not sufficient, the morphology of the Si electrode can change significantly owing to the inherent volume change of Si, as evidenced by the large cracks observed in PAA/SiNPs.

To investigate the physical stability of PAA/SiNP and W-PI-30/SiNP quantitatively, the adhesion strengths of both Si electrodes were investigated using SAICAS [45]. In these tests, the blade was placed between the Si electrode composite and Cu current collector, and the adhesion strength was evaluated by moving the blade horizontally. The adhesion strength of W-PI-30/SiNP (0.296 kN m^−1^) was around double that of PAA/SiNP (0.146 kN m^−1^) (Figure 6). Thus, it was inferred that the excellent adhesion between W-PI-30 and SiNPs played an important role in inhibiting morphological changes and improving the cycling performance of W-PI-30/SiNP.

## 4. Conclusions

We synthesized W-PAmAS as a water-soluble PI precursor using a facile one-step process for use as a Si anode binder in LIBs. DABA was added to W-PI to introduce carboxylic acid moieties and increase the binding force by forming covalent bonds with the SiNPs. This interaction significantly reduced the volume changes of the SiNPs during the alloying/dealloying process. In addition, because an imidization degree of >90% was achieved when W-PAmAS was annealed at 150 °C, the low-temperature process demonstrated in this work can effectively suppress the unwanted oxidation of the Cu current collector during electrode manufacturing for LIBs. The cells with Si electrodes using W-PI-30 as a binder showed superior cycling stability with 91.3% of initial discharge capacity maintained after 200 cycles and high rate capability even at a high current density of 6.0 A g^−1^. Furthermore, the adhesion strength was almost double that of the conventional PAA binder. These remarkable performances are attributed to the inhibition of the morphological changes of SiNPs by improved chemical bonding, as well as the robust mechanical properties of the W-PI binder. Therefore, W-PI-based polymeric binders have great potential for use as novel water-soluble binders in Si anodes with suitable electrochemical performance for LIBs.

## Figures and Tables

**Figure 1 nanomaterials-11-03164-f001:**
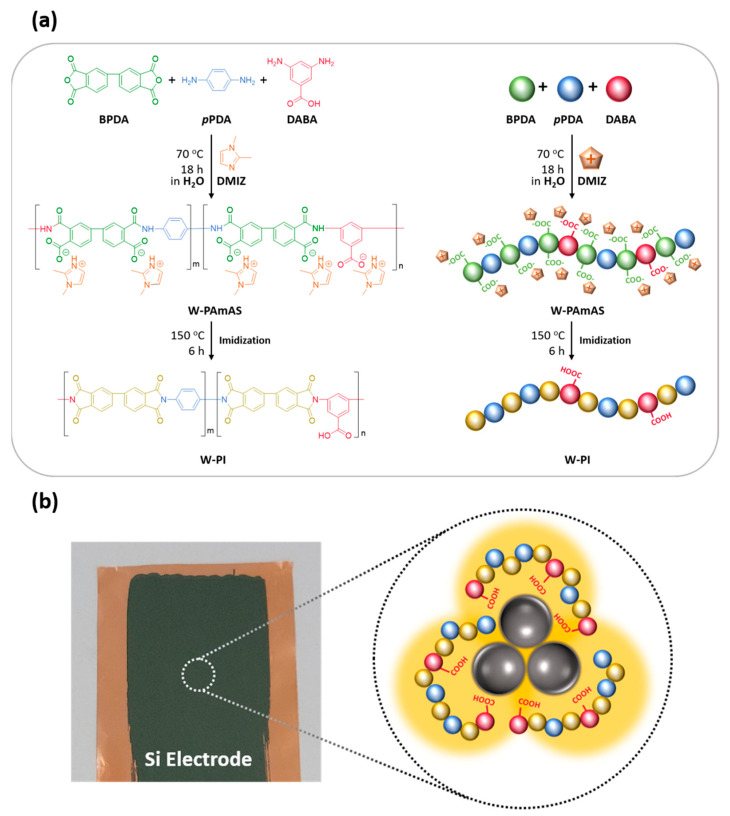
(**a**) Synthesis scheme of W-PAmAS-30 and preparation of W-PI-30. (**b**) Photograph of the Si electrode on Cu foil fabricated using W-PI-30 as a binder after thermal treatment at 150 °C for 6 h and schematic illustration of chemical interactions between SiNPs and the W-PI-30 binder.

**Figure 2 nanomaterials-11-03164-f002:**
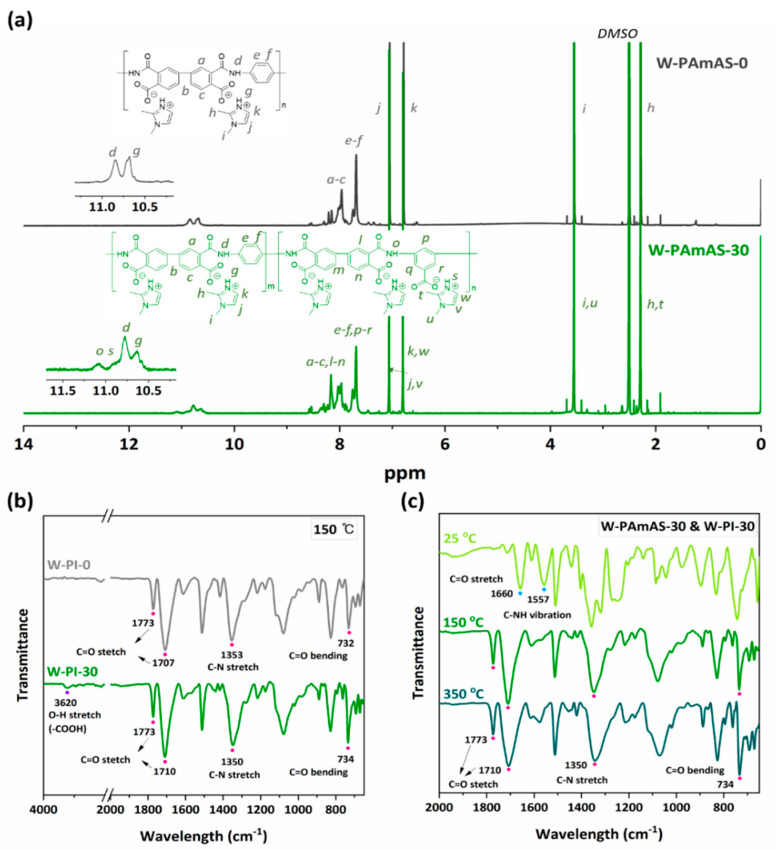
(**a**) ^1^H NMR spectra of W−PAmAS−0 and W−PAmAS−30 in DMSO−d_6_. FT-IR spectra of (**b**) W−PI−0 and W−PI−30 thermally annealed at 150 °C for 6 h and (**c**) the effects of heat treatment temperature on the imidization of W−PAmAS−30.

**Figure 3 nanomaterials-11-03164-f003:**
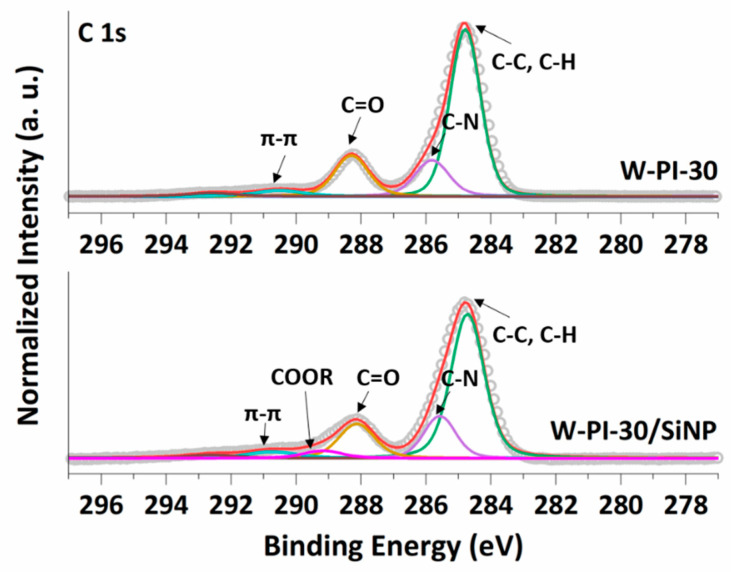
XPS C 1s spectra of W-PI-30 and W-PI-30/SiNP.

**Figure 4 nanomaterials-11-03164-f004:**
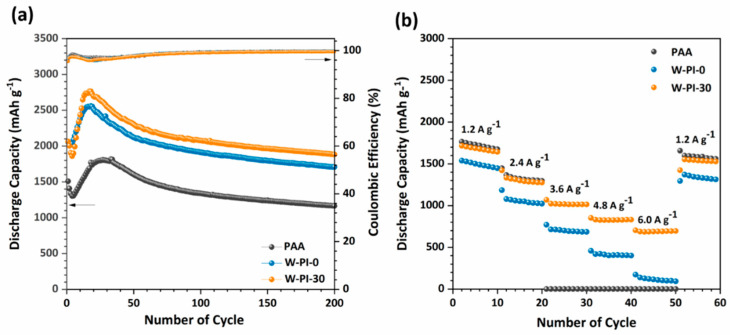
(**a**) Cycling performance and (**b**) rate capabilities of Si electrodes with PAA, W−PI−0, and W−PI−30 binder systems. Experimental conditions: (**a**) 200 cycles, current density: 1200 mA g^−1^, potential range: 0.05−2.0 V at 25 °C.

**Figure 5 nanomaterials-11-03164-f005:**
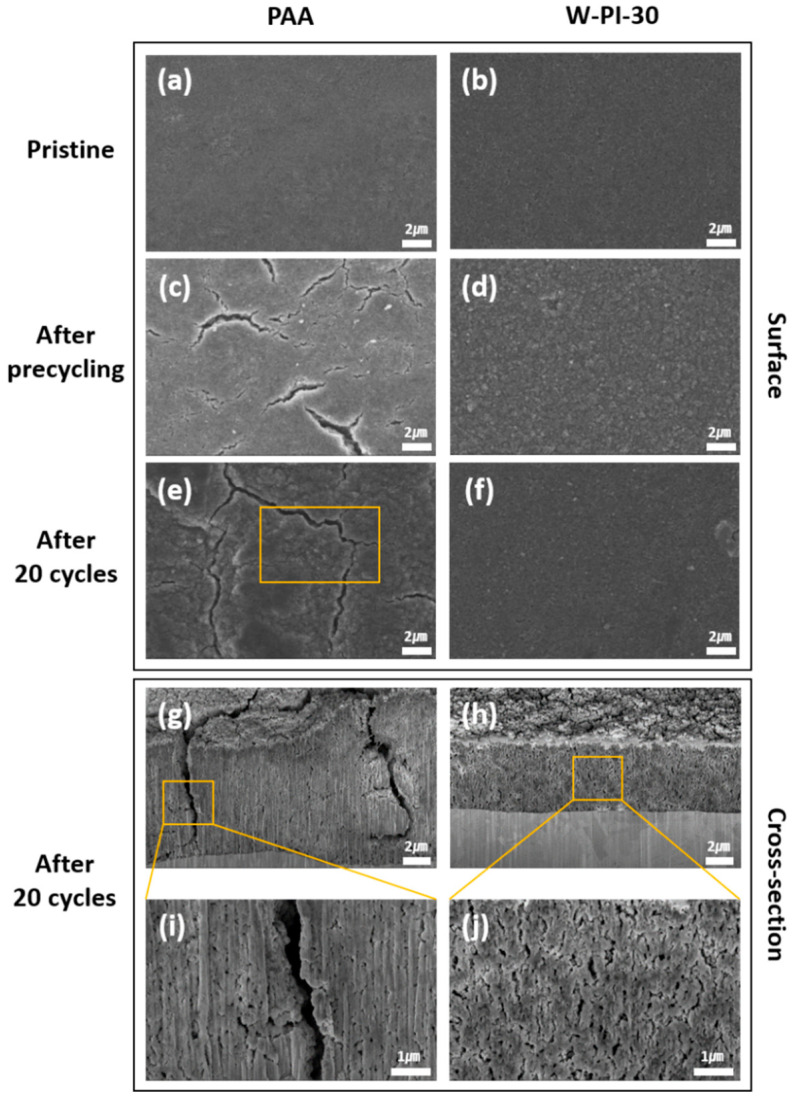
FE-SEM images of Si electrodes with PAA and W-PI-30 binders. (**a**–**f**) Surface morphology of Si electrodes before cycling, after precycling, and after 20 cycles ((**a**,**c**,**e**) PAA and (**b**,**d**,**f**) W-PI-30). (**g**–**j**) Cross-sectional images of Si electrodes with PAA and W-PI-30 binders after 20 cycles.

**Figure 6 nanomaterials-11-03164-f006:**
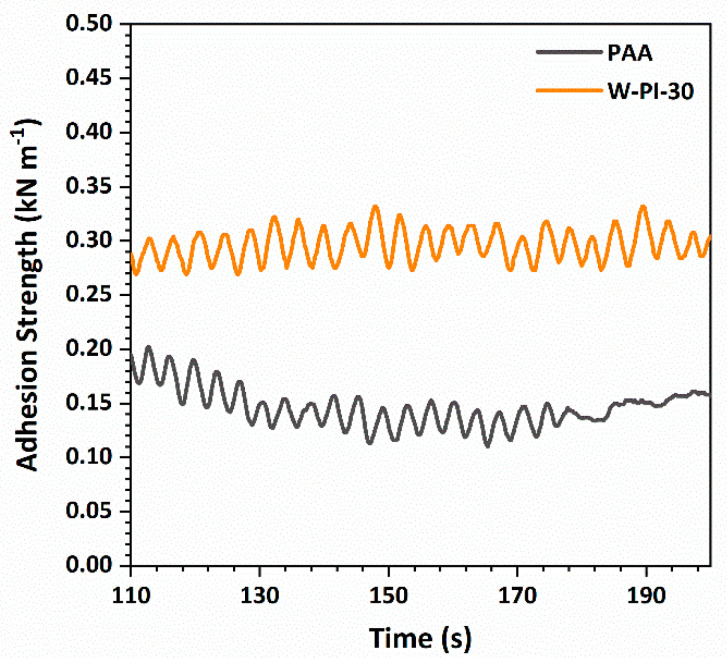
Adhesion strength of Si electrodes with PAA and W−PI−30 binders measured using SAICAS.

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
