# Peer review of "Eco-Friendly Water-Processable Polyimide Binders with High Adhesion to Silicon Anodes for Lithium-Ion Batteries"

_nanomaterials, 2021, doi:10.3390/nano11123164_

Round 1
Reviewer 1 Report
W-PAmAS as a water-soluble PI precursor was synthesized by a one-step process for use as a Si anode binder in LIBs. DABA was added to W-PI to introduce car-boxylic acid moieties and increase the binding force by forming covalent bonds with the SiNPs. The as-prepared Si-based anode showed improved electrochemical-cycling stability. The results are interesting. The following points need to be addressed.
- In Fig. 4a, the capacity first increases and then gradually decreases with the electrochemical cycling. This needs to be reasonably explained.
- Initial discharge/charge curves should be given to provide more electrochemical properties.
- Electrochemical impedance spectra (EIS) are suggested to be performed for achieving the electrode kinetics, which is beneficial to understand the effects of polyimide binders.
- ‘The Si anode with W-PI binder showed improved electrochemical performance with a high capacity of 2061 mAh g-1 and excellent cyclability of 1883 mAh g-1 after 200 cycles’ is stated in the abstract section. However, the rate must be provided.
Author Response
There are several pictures and various references for responses, so I submit it as a file.

Reviewer 2 Report
The manuscript reported the synthesis of water-processable polyimide Binders for silicon anodes, and the electrochemical tests indicated that the binders could significantly improve the cycling stability of the corresponding electrodes. The manuscript could be accepted for publication in Nanomaterials after the following revisions.
1. In Figure 4, the Si electrodes with W-PI-0 and W-PI-30 binders exhibited about higher capacities of 2000-2500 mAh g-1, and with PAA exhibited capacities lower than 1500 mAh g-1 at 1.2 A g-1 for the first 10 cycles during the cycling tests. However, the rate capability tests did show large different capacities and trends at the same current density for the first 10 cycles, i.e. the Si electrodes with PAA and W-PI-30 PAA exhibited similar capacities of around 1700 mAh g-1, whereas that with W-PI-0 exhibited lower capacities of around 1500 mAh g-1. Therefore, the authors need to retest their samples to provide more accurate results.
2. EIS measurement with an appropriate discussion is recommended so as to get a further understanding about the influence of the binders on the electrochemical performance of the silicon electrodes.
Author Response

(The authors gave the same response as above.)

Reviewer 3 Report
The article “Eco-friendly water-processable polyimide binders with high adhesion to silicone anodes for lithium-ion batteries“ is fitting the Nanomaterials scope, is written in good English and is well organized.
There are some important points that should be addressed:
1) The idea of using water-soluble binders for lithium-ion battery silicon anodes is not new and polyimide is not quite an exception. There is even a commercial binder of this kind (DREAMBOND, https://www.istusa.com/industrial_material/dreambond/).The novelty of this paper is questionable, the authors should emphasize this aspect in the Introduction to convince their work is novel and is worth publishing.
2) Comparison with literature of the performance of their anode is missing and is necessary.
3) Did the authors performed impedance spectroscopy to determine the capacity of their anode? This is one point that is also normally addressed when this kind of electrode is developed.
Small errors occurred, that should be also addressed in order to improve the manuscript, some suggestions are given bellow:
Page 3, Line 119: Please put the temperature in increasing order to be easier to follow.
Page 4, Line 150: Please add “temperatures” after “different” or remove “different” to be more correct.
Page 8-9: Try to maintain Figure 4 and its caption on the same page.
Page 9-10: Try to maintain Figure 5 and its caption on the same page, especially because Figure is presented before text discussion.
Author Response
There are several pictures and various references for responses, so I submit it as a file. Please see the attachment.

Round 2
Reviewer 1 Report
The previously raised points have been elucidated or addressed in the revised version. It may be accept for the publication.
Reviewer 2 Report
The quality of the manuscript has been improved, and it can be accepted for publication.
Reviewer 3 Report
The authors addressed all the questions and improved their manuscript. I recommend publication in the present form.